# The Importance of Nanocarrier Design and Composition for an Efficient Nanoparticle-Mediated Transdermal Vaccination

**DOI:** 10.3390/vaccines9121420

**Published:** 2021-12-01

**Authors:** Rayen Yanara Valdivia-Olivares, Maria Rodriguez-Fernandez, María Javiera Álvarez-Figueroa, Alexis M. Kalergis, José Vicente González-Aramundiz

**Affiliations:** 1Departamento de Farmacia, Escuela de Química y Farmacia, Facultad de Química y de Farmacia, Pontificia Universidad Católica de Chile, Santiago 7820436, Chile; ryvaldivia@uc.cl (R.Y.V.-O.); mjalvare@uc.cl (M.J.Á.-F.); 2Institute for Biological and Medical Engineering, Schools of Engineering, Medicine and Biological Sciences, Pontificia Universidad Católica de Chile, Santiago 7820436, Chile; marodriguezf@uc.cl; 3Millennium Institute on Immunology and Immunotherapy, Departamento de Genética Molecular y Microbiología, Facultad de Ciencias Biológicas, Pontificia Universidad Católica de Chile, Av. Libertador Bernardo O’Higgins No. 340, Santiago 7810000, Chile; 4Departamento de Endocrinología, Escuela de Medicina, Pontificia Universidad Católica de Chile, Santiago 7810000, Chile; 5Millennium Institute on Immunology and Immunotherapy, Departamento de Farmacia, Escuela de Química y Farmacia, Facultad de Química y de Farmacia, Pontificia Universidad Católica de Chile, Santiago 7820436, Chile; 6Centro de Investigación en Nanotecnología y Materiales Avanzados “CIEN-UC”, Pontificia Universidad Católica de Chile, Santiago 7810000, Chile

**Keywords:** transdermal vaccines, needle-free immunization, nanomedicine, nanoparticle design, nano vaccines

## Abstract

The World Health Organization estimates that the pandemic caused by the SARS-CoV-2 virus claimed more than 3 million lives in 2020 alone. This situation has highlighted the importance of vaccination programs and the urgency of working on new technologies that allow an efficient, safe, and effective immunization. From this perspective, nanomedicine has provided novel tools for the design of the new generation of vaccines. Among the challenges of the new vaccine generations is the search for alternative routes of antigen delivery due to costs, risks, need for trained personnel, and low acceptance in the population associated with the parenteral route. Along these lines, transdermal immunization has been raised as a promising alternative for antigen delivery and vaccination based on a large absorption surface and an abundance of immune system cells. These features contribute to a high barrier capacity and high immunological efficiency for transdermal immunization. However, the stratum corneum barrier constitutes a significant challenge for generating new pharmaceutical forms for transdermal antigen delivery. This review addresses the biological bases for transdermal immunomodulation and the technological advances in the field of nanomedicine, from the passage of antigens facilitated by devices to cross the stratum corneum, to the design of nanosystems, with an emphasis on the importance of design and composition towards the new generation of needle-free nanometric transdermal systems.

## 1. Introduction

The recent pandemic caused by the SARS-CoV2 infection has shown the great importance of vaccines and their impact on preventing and controlling infectious diseases [1]. As a result, the attention to developing safe and effective vaccines has increased. However, the main route of antigen delivery remains parenteral, reducing the possibility of universal coverage since it can be considered traumatic for some individuals, requires qualified health professionals for application, and, in many cases, efficient cold chain management [2,3]. These factors make it challenging to develop vaccination programs in developing countries or remote areas [4,5]. In addition, the World Health Organization (WHO) estimates that at least 19.4 million infants in the world have not received basic vaccines, a scenario that has worsened due to the pandemic. Therefore, the need to find alternative routes for efficient, safe, and effective antigen delivery to induce protective immunity [6].

The transdermal route of administration provides multiple advantages for achieving this goal because it reduces both the first-pass metabolism and adverse effects, is non-traumatic, and allows self-administration by the patient making it an attractive delivery route for needle-free immunization [7,8]. However, the protective barrier function of the skin can restrain the step of macromolecule and antigen absorption. Those that manage to overcome the stratum corneum (SC) may be available to exert their pharmacological effect [9,10].

The skin is an easily accessible and highly immunocompetent organ [7,11], which can have up to 20 billion cells of various subtypes, such as keratinocytes, Langerhans cells, dendritic cells, T cells, and mast cells that contribute to the immunocompetence of skin [12]. To overcome the barrier that the stratum corneum imposes and favor transdermal permeability, various technologies have been developed, including iontophoresis [13], sonophoresis [14], magnetophoresis [15], electroporation [16], and laser microporation [17]. Unfortunately, these methods have shown significant economic limitations [18]. Therefore, microneedles are the most widely studied method to administer micro and macromolecules through the skin [19]. However, many researchers do not consider this delivery method as a “needle-free” approach. Thus, the development of highly efficient and optimized nanosystems is one of the strategies to cross the skin barrier and benefit from the immunocompetence of this tissue, with design being one of the factors to consider when penetrating the skin without invasive techniques. A solution to this problem could be developing nanosystems capable of transporting the antigen and bypassing this barrier without microneedles. Fulfilling this last objective largely depends on the design composition of the nanoparticles [20]. This article will focus on the advances that have been achieved in the area, with a strong focus on the design techniques of nanoparticle-based transdermal antigen delivery systems and the role of the configuration and use of excipients that favor the crossing of the skin barrier.

## 2. Mechanisms Involved in the Skin Immune Response

The skin is an extensive and complex organ that accomplishes a fundamental barrier function and comprises various layers that develop in different stages of gestation [21]. The epidermis development is a complex but coordinated process involving cell proliferation, differentiation, and adhesion steps [22]. This process begins in the first weeks of embryonic development, and stratification extends until the end of the first trimester of embryonic development, which ends with the differentiation of spiny cells into granular and cornified cells [22]. The dermis is organized in more advanced stages and continues its maturation weeks after birth [23,24]. Although cells of the immune system in the skin are not usually so abundant, a great density and diversity of immune cells are achieved after a complex development process, which constitutes the skin as a specialized barrier organ [25].

The properties of this barrier are granted mainly by the presence of the stratum corneum, which consists of the outermost layer of the skin, located on the viable epidermis with a thickness of around 15 to 20 layers composed mainly of dead tissue, assuming a barrier almost impenetrable for the vast majority of molecules with therapeutic activity [9]. But not only the intrinsic properties of SC can influence the transdermal passage, but also the physicochemical properties of the compounds can define entry efficiency. The physicochemical properties include molecular weight, solubility, and lipophilicity, which define their ability to be absorbed. Molecules with low molecular weight (less than 500 Da) [26] and log P between 1–4 are expected to diffuse easily [27]. On the other hand, larger molecules but with sufficient lipophilicity could enter through the annexed pathways that we will explain later. All these skin characteristics pose challenging work in administering assets via the transdermal route [8].

The passage of bioactive compounds through the stratum corneum is the first critical point of interest for delivering drugs through the skin. As mentioned above, the specific physicochemical characteristics are restricted and, if it is sufficiently lipophilic, it can enter through the lipophilic stratum corneum [26]. The passage of compounds through the skin can occur through the transepidermal pathway, either through the transcellular pathway that involves the passing through SC cells or the intercellular pathway (also known as paracellular) through the spaces between corneocytes (approximately 75 nm) [28]. On the other hand, the transpedicular pathway consists of skin attachments, such as sweat glands, sebaceous glands, and hair follicles. Normally, this route is not relevant for drug administration as it constitutes only 0.1% of the human skin [29]. However, it plays an important role in highly lipophilic drugs that could form reservoirs in the sebaceous glands or high molecular weight compounds such as nanoparticles, facilitating their entry through hair follicles. Figure 1 shows the various access routes of the active ingredients through the skin [30,31].

Once the molecule enters through the stratum corneum, it will face the interface between the viable epidermis and the dermis with hydrophilic characteristics; given the above, only compounds capable of ionizing will cross, facing the enzymatic activity of the skin [32]. As we can see, the passage of molecules through the skin poses significant challenges. In this regard, various approaches have been developed to facilitate the passage of active compounds. Specifically, in this review, we will address advances in immunization since it is in this field where we find advantageous characteristics for administration.

Drug or antigen delivery via the transdermal route has several advantages (Figure 2). First, pre-systemic metabolism can be avoided, being a candidate for compounds with extensive hepatic metabolism. Second, the transdermal route offers a minimally invasive approach. Third, transdermal delivery avoids infections associated with the manipulation of conventional needles. Fourth, the transdermal route allows dose reduction due to the existence of minimal metabolism. Finally, it has the potential for self-administration and effective induction of the immune system, making it an attractive route for non-invasive immunization [30,31].

### Innate and Adaptive Immune Response of the Skin

The potential of transdermal immunization is supported by the abundant presence of cells of the immune system in the skin, which can trigger an effective antigen-specific immune response [33]. Langerhans cells were initially described in 1868 by Paul Langerhans [34] and correspond to a subtype of dendritic cells with a stellate shape located mainly at the base of the epidermis [35]. Langerin expression characterizes these cells; this protein plays a fundamental role in presenting antigens to T25 cells and corresponds to a type C lectin, which is localized in cytoplasmic organelles with a striated appearance inside Langerhans cells called Birbeck granules [36]. It has been recently shown that the expression of the kinases activated by serine/threonine p21 (PAK1) in Langerhans cells contributes significantly to the maintenance of epidermal stem cells, which in turn can be related to autoimmune pathologies and skin cancer, underscoring the importance of this cell type for skin immunomodulation [37]. An essential link between innate and adaptive immunity is the dendritic cell, which can activate naïve T cells and contributes to the initiation of both cellular and adaptive humoral immunity [38], Specifically, in the skin, we can find dermal dendritic cells, which correspond to a subtype of dendritic cells; the evidence shows that they present a greater activation than blood dendritic cells, promoting a strong proliferation of T cells, two populations of dermal dendritic cells; CD1c + DC and CD141 + CD, the latter being responsible for the cross-presentation of CD8 + T antigens [39]. On the other hand, the recent discovery of various subtypes of innate lymphoid cells (ILC), such as ILC1/2/and 3, has contributed to the complexity of the immunomodulation mechanisms in the skin [40,41]. Despite coming from a common lymphoid progenitor, ILCs lack the specific rearranged antigen receptors expressed by T cells and the three ILC subtypes located at different skin layers [42].

Keratinocytes (KCs) are cells that produce keratin in the skin and constitute a high percentage of the cells of the epidermis response [39], forming an efficient barrier, which works as the first line of defense against skin pathogens and exogenous substances [43]. KCs express Toll-Like receptors (TLR) and secrete several types of chemokines and proinflammatory cytokines in response to TLR stimulation by PAMPs. Thus, high expression of interleukin 33 (IL-33), a member of the IL-1 family, has been shown to activate helper T cells, macrophages and induce the ILC innate lymphoid cell family [44]. The recent discovery of various subtypes of innate lymphoid cells (ILC), such as ILC1/2/and 3, has added to the complexity of the immunomodulation mechanisms in the skin [40,41]. Despite coming from a common lymphoid progenitor, ILCs lack the specific rearranged antigen receptors expressed by T cells and the three ILC subtypes located at different skin layers [42]. Dermal fibroblasts are another cell type that makes up the skin and express TLR-type receptors even at higher levels than keratinocytes. One of the functions of these cells is to secrete components of the extracellular matrix [45].

## 3. Transdermal Immunization Based on Physical Methods to Go across the Stratum Corneum

As discussed above, a significant challenge for administering transdermal vaccines is to overcome the stratum corneum [46]. In this regard, various approaches have been explored, based either on devices disrupting the skin barrier or on vehicles that facilitate antigen passage through the skin. Figure 3 summarizes the most commonly used approaches to achieve this transdermal penetration. These approaches have been used alone or in combination with nanosystems, described in the following sections.

### 3.1. Transdermal Administration Based on Microneedles

One of the strategies that have been successfully applied for transdermal administration and that causes minor disruption of the stratum corneum is based on microneedles [47]. This strategy manages to overcome the limitations of the conventional parenteral route, as needles with sizes of micrometers can penetrate the layers of the skin, avoiding the typical discomfort in the patient due to pain, bleeding, and risk of infections [48]. Furthermore, microneedles do not suffer from limitations associated with the particle size of bioactive compounds since these devices usually have a larger size. Because of these features, there are already several products marketed for therapies, such as the intradermal influenza vaccine Intanza^®^ [49] or the acne treatment Dermaroller [50]. Furthermore, microneedles are usually classified according to the material with which they have been produced or the active release profile. Thus, we can find dissolving-, solid-, coated- and hollow-microneedles [51].

In the field of transdermal immunization, microneedles associated with free antigen or combined with nanosystems have been used successfully. Table 1 shows some nanosystems designs for diphtheria and a new DNA vaccine against SARS-CoV-2. Microneedles are undoubtedly the most developed devices to date in transdermal immunization [52]. Although recently a very low-cost microneedle-anchored electroporator device for immunization against SARS-CoV-2 was shown, there are still challenges to overcome [53]. Current studies aim to overcome the remaining challenges in this area [54]. New technologies have been developed to manufacture microneedles to obtain improved compatibility when entering the stratum corneum. Examples are soluble microneedles, made with totally biocompatible water-soluble materials that penetrate the skin barrier and then solubilize with the active principle [55]. The oxidation of the material and the shortening of the microneedle length to avoid skin irritation has been another focus of study, leading, for example, to the development of microporous polymeric microneedles [56]. The biocompatibility of the devices has also been an essential subject of study, due to the presence of possible unwanted effects in the application site, such as irritation, inflammation, pore enlargement, and modification of the skin barrier [57,58]. It should be noted that these challenges to be overcome are common to other administration methods; this review will not focus on microneedles mainly; however, we do recommend to the reader some excellent reviews to delve into the advances of this technology [59].

### 3.2. Transdermal Administration Based on Electrical Techniques

Iontophoresis consists of the application of electrical current through the skin to favor the penetration of specific molecules. This method is an effective and non-invasive route of penetration [67]. Among the limitations that this administration technique faces for transdermal vaccination is the difficulty of administering the antigen in a focused way, avoiding permeation towards the muscle, and efficiently achieving its accumulation and subsequent stimulation of Langerhans cells. Combining this technique with nanoencapsulated bioactive compounds is a promising approach to overcoming this challenge [68,69].

Other techniques that use this basis for the delivery of bioactive compounds include sonophoresis, in which ultrasonic energy is used to induce the entry of assets through the skin [70]. Another technique is magnetophoresis, in which electric charges induce a magnetic field and with this occurs the vectorization of the drug [71]. We also have electroporation; in this technique, an aqueous pore is created in the skin by exposing it to high voltages for short periods, allowing the entry of bioactive compounds [72]. Finally, we find microporation, which, like electroporation, is based on creating a pore that will enable the passage of bioactive compounds. Still, this time the energy is transmitted through a metallic element by conduction, which produces a non-transitory pore in the skin due to the increase in temperature at the level of the stratum corneum [73]. Figure 3 shows a schematic representation of the active diffusion techniques and passive diffusion through barriers. It is important to note that once the stratum corneum has been crossed, the particles can release their content to produce the immune response.

### 3.3. Transdermal Administration Based on Other Approaches

Star-shaped particles have been developed, made of aluminum oxide or stainless steel, which can generate pores in the skin and thus overcome the stratum corneum barrier. The authors achieved surprising results in improving the survival of mice with cutaneous melanoma treated with 5-fluorouracil and in vaccination against tetanus toxin [74]. However, it is still unclear whether these pores are harmful to the skin in the long term. That is why in the next section, we will review the approaches that have been made to achieve a needle-free vaccination, focusing on the design and composition.

Another relatively novel device used is the PharmaJet Needle-Free Jet injector device, which has been shown to successfully administer the influenza virus vaccine in a randomized trial that compared it with intramuscular administration. The results were not inferior to those obtained with conventional administration, offering an alternative to traditional needle syringes since the device does not have a needle [75]. However, the costs associated with this type of device remain a disadvantage in this field. Table 1 shows another example of the successful use of this device to administer the ovalbumin antigen. Additionally, it incorporates the recent advances in nanosystem technologies used for immunization in in vivo tests for different antigens.

## 4. Nanosystem-Based Antigen Delivery Systems Noninvasive; Needle-Free Administration

Until now, we have reviewed the characteristics that make the skin and the transdermal pathway promising for immunization and the administration techniques used to cross the stratum corneum and their limitations, highlighting as the main challenge the achievement of immunization without damaging the stratum corneum.

One of the factors that make needle-free immunization desirable is the fact that the rupture methods used to allow the passage of particles such as microneedles, microporation, abrasion, among others, not only produce the response of the innate immune system, generating skin reactions adverse effects in patients, but also interrupt the skin barrier, which implies a greater risk of infections. This does not mean a greater compromise in healthy patients; however, in immunocompromised people, patients with difficult healing, children, and the elderly pose a greater risk that limits extensive use, without also considering the use of these devices complicates administration [76].

In this race to obtain adequate and safe transdermal vaccines, the incorporation of nanomedicine has made a significant contribution [77]. This has allowed, for example, to improve the revised administration techniques with the incorporation of controlled antigen release systems that have allowed to overcome some limitations such as thermostability [78], to enhance permeability by having a small particle size and increasing the contact surface [79], to reduce doses avoiding the manifestation of adverse reactions and to improve pharmacokinetic profiles [80]. The main advantages of nanovaccines are summarized in Figure 4. Next, this review will address the advances in administering vaccines through the skin using nanosystems, their types, designs, approaches, and challenges for the design and composition of nanosystems.

The most widely used nanosystems for antigen delivery are nanoparticles, liposomes, polymeric nanoparticles, niosomes, cubosomes, ethosomes, gold nanoparticles, and nanoemulsions. The choice of the type of nanometric particle depends mainly on the bioactive characteristics of the compound and the chosen route of administration; therefore, it is crucial to consider its use either combined with some administration method of those already reviewed or by itself. The design orientation is based on the delivery of passive or active bioactive compounds. Table 2 summarizes the various types of nanosystems used for antigen administration, their properties, their specific application in the immunization area, and the main challenges that remain to be faced with making their use in the field of immunization. Next, we will address some nanosystems that have been more widely used.

### 4.1. Liposomes

Liposomes correspond to a double layer commonly formed by phospholipids or other derivatives and cholesterol [95]. These systems have been widely studied to transport antigens and active molecules due to their possible adjuvant effect, triggering an efficient immune response [96,97]. On the other hand, their composition gives them high biocompatibility and the possibility of directing immunological therapies to the different targets in a controlled way [88,98]. The properties that affect this process are the physicochemical characteristics such as particle size, Z potential, polydispersity, and lipid composition [99,100,101]. To date, multiple approaches have been developed to combat infectious diseases by associating antigens with this type of nanocomposite [102,103,104]. Its efficacy for administering antigens against SARS-CoV-2 by different routes of administration is still being evaluated [105]. In the area of transdermal immunization, they have been used in conjunction with dissolvable microneedles to develop vaccines against leishmaniasis [106] and for non-invasive delivery of vaccines against tetanus toxoid [107], among others (see Table 2).

### 4.2. Nanocomposites Derived to Liposomes

In the field of needle-free transdermal immunization, it has been suggested that the rigid structure of liposomes makes it challenging to pass through the skin barrier [89], which is why multiple modifications have been incorporated to create liposome-derived nanosystems that can circumvent such limitations.

#### 4.2.1. Transferosomes

Transferosomes are elastic liposomes composed of phospholipids, which form deformable vesicles and increase transdermal permeability in the presence of a hydration gradient in the stratum corneum [108]. Their composition, based mainly on edge activating surface surfactants such as sodium cholate, polysorbates, and Sorbites, allows a modulation in the flexibility of the sheath, allowing them to pass through the pores of the skin, thus opening the way to needle-free vaccination [109]. Among the advantages of using these designs are their high flexibility, their ability to encapsulate hydrophilic and hydrophobic compounds, and their ability to incorporate molecules of peptide origin [110]. Due to their composition, they lack biocompatibility problems; like other nanosystems and can be used for topical and systemic treatments.

The mechanism of transdermal entry of transferosomes is based on the presence of border activators, in a first step; They allow the nanosystem to pass through the stratum corneum through channels with diameters of less than 50 nm. In a second step, the component derived from phospholipids is capable of sealing the vesicle and transporting it through the pore, the gradient produced by the difference in water content between the surface of the skin and the epidermis produces what is known as “transdermal gradient”, which allows the passage of the nanosystem [111]. This is how they are capable of crossing the skin barrier by the transcellular or intercellular route [112].

Despite its beneficial properties for transdermal immunization, there is currently a small number of studies in this field, we can mention tetanus vaccine designs [113] and against the virus responsible for hepatitis B [86]. The above is a motivation to overcome the challenges that limit its use, mainly derived from its oxidative stability [114], high cost associated with the constituent lipids, and difficulty of reproducibility of the preparations [115]. So, there is still a long way to understand the interaction between the compounds, components, and stabilization of these nanosystems to develop transdermal vaccines.

#### 4.2.2. Ethosomes

Ethosomes correspond to a type of liposome that contains in its composition between 20 to 40% ethanol. Their properties include a great drug encapsulation capacity and stability in comparison with the classic liposomes, negative Z potential, size smaller than 200 nm, which decrease as the ethanol concentration increases, highly deformable, non-toxic and highly biocompatible [116], skin permeability facilitated by ethanolic content [117,118]. The passage mechanism through the stratum corneum of ethosomes involves two steps: the contact of SC lipids with ethanol produces a composition alteration, known as the “ethanol effect”. In a second phase, the breakdown of the superficial lipids generates a decrease in the skin barrier compaction that allows the flexible structure of ethosomes to enter through the skin and interact with the polar component of lipids, known as the “ethosome effect” [119]. By using ethosomes marked with fluorescent probes, the molecular mechanism of entry of ethosomes was alucidated. Thus, the passage through the SC by intercellular pathways can take place without damaging the structure of the SC and distributing mainly in the cell membrane.

Ethosomes have been developed for various applications, such as treatment of inflammation, analgesia, skin conditions, among others [120]. Although few studies have been carried out in the field of transdermal immunization, it has been shown enhanced transdermal penetration by ethosomes marked with rhodamine [121]. Additionally, ethosomes in conjunction with biopolymers for the administration of ovalbumin, effectively stimulating the response of the immune system and they have been incorporated into carbomer gels to achieve vehicles that can be administered needle-free [122]. It was also shown that ethosomes that include hyaluronic acid or chitosan formulated by layer-by-layer self-assembly promote the stimulation of IL-2 and IL-6 and cytokines associated with dendritic cell maturation when loaded with an antigen [89]. The challenges associated with the formulation of this type of nanosystem are summarized in Table 2.

#### 4.2.3. Niosomes

Niosomes are liposome-type nanometric structures formed by the assembly of non-ionic surfactants, which are mainly derivatives of alkyl or dialkyl polyglycerol ether and cholesterol that are subsequently hydrated. Since they were discovered in the cosmetic industry in the 1980s, these structures have been widely formulated. Depending on their manufacturing method, Niosomes can be unilamellar or multilamellar (obtained by thin layer evaporation, addition of molten lipids, addition of hot water, ether injection, microfluidics, among others) [123]. Advantages of using niosomes include easy large-scale production, high stability, low toxicity, and high transdermal penetration [124].

A successful example of the use of niosomes for transdermal vaccination is a 60% increase in the immune response by encapsulating the antigen to prevent Newcastle disease [125]. However, in recent years, despite the great apogee of niosomes, no significant advances have been reported in the field of transdermal immunization. This scenario could be due to the use of organic solvents, phase heating (non-appropriate for thermolabile compounds), or a large number of existing patents for these vehicles.

#### 4.2.4. Cubosomes

Cubosomes are composed of two aqueous channels, separated by a lipid bilayer arranged three-dimensionally either in a lipid form, water, double diamond, rotated, or primitive [126,127]. Among the lipids most used to formulate them is the [128], while surfactants derived from poloxamers have been the most used to stabilize the cubic structures of these nanosystems [129]. Among the ways of obtaining it, the bottom-up approach and top-down approach stand out. The latter is the most used and consists of the formation of the viscous primary structure formed by lipids and its subsequent dispersion in water by applying high-energy methods [130,131]. The designs for transdermal immunization of niosomes and cubosomes are still small (Table 2) and are not without challenges when formulating them. It is expected that with the progress of research and incorporation of suitable stabilizers, these limitations can be overcome by the encapsulation of assets and transdermal penetration, preventing them from losing their conformation as they pass through the various layers of the skin [132].

### 4.3. Nanoparticles

Nanoparticles correspond to colloidal structures of nanometric size, for nanomedicine approaches, usually less than 500 nm. Depending on their formulation, we can find polymeric nanoparticles associated with polymers by interacting electrostatic charges or nanocapsules, which generally have a lipid core and a polymer shell. Both designs can incorporate compounds of therapeutic interest either by encapsulation or association by adsorption on the surface [80,133,134].

#### 4.3.1. Polymeric Nanoparticles

New approaches based on incorporating biodegradable polymers into their composition have shown great potential in the biomedical area and the field of immunization [135]. These nanosystems are capable of containing not only antigens of interest but also various adjuvants. Being biodegradable, they can maintain the release of compounds from days to several weeks generating biocompatible waste [81,82]. Table 2 shows recent examples of formulations for transdermal immunization using these nanocarriers.

The pandemic caused by the SARS-CoV-2 virus has not only triggered a broad race to find effective vaccines against this virus. Still, it has also allowed novel vaccines based on nanoparticles loaded with messenger RNA to reach the market, positioning itself as the first of such designs to be approved by regulatory agencies such as the EMA and the FDA [136]. This is how vaccines such as Pfizer and BioNTech RNA: BNT162b2 and modern mRNA-1273 based on purified messenger RNA are used today [137,138]. Although these designs are administered parenterally, without a doubt, they open the door to the development of new technologies, aiming at needle-free vaccination and is not only limited to the transport of antigens but also undoubtedly an open door towards the development of new therapies for the treatment of COVID-19 [139].

#### 4.3.2. Nanocapsules

Systems with nanometric size and core-shell structure and coated mainly with polymeric compounds are called nanocapsules. Among their characteristics, we find the ability to induce long-lasting immune responses, dose reduction, and reduction of adverse effects [140]. Our research group has worked on the development of nanocapsules capable of carrying antigens for needle-free transdermal immunization and shown that polymeric nanocapsules with chitosan shell, loaded with Ovalbumin (OVA) are stable, with a high association of the protein capable of interacting with the immune system and being, in an ex vivo model in pigskin, better retained than OVA in solution [33]. This property has been reinforced by associating hyaluronic acid as a biopolymer, showing promising systems for needle-free transdermal administration [141].

### 4.4. Nanoemulsions

Nanoemulsions are heterogeneous mixtures that can be formed by drops of oil in an aqueous medium (O/W) or drops of water dispersed in oil (W/O), stabilized by incorporating surfactants and being of nanometric size [60]. They have shown great potential in transdermal vaccination; however, it is still necessary to study the mechanisms by which they could cross the skin barrier. Until now, there is a history that this is dependent on size, which is mainly called the note is that in this case, smaller sizes do not necessarily imply a greater transdermal passage [142].

In the case of transdermal immunization, they have been designed in conjunction with Imiquimod to induce enhanced responses in T lymphocytes [143]. In the case of the incorporation of biopolymers, the influence of the polymeric coating on transdermal penetration has been studied [144]. Recently, the nanoemulsion MF59, an authorized and approved preparation with commercial use for the administration of parenteral vaccines against influenza, has been associated with microneedles, demonstrating a painless administration and maturation of dendritic cells [145]. Although only a few studies for the use of nanoemulsions for transdermal vaccination have been published, their versatility, stability, and a large number of studies associated with their excellent safety profile make them promising vehicles for the fight against COVID-19 disease, not only as antigen carriers but also as carriers of various active molecules against the SARS-CoV-2 virus [146].

## 5. Novel Approaches to Design Nanoparticles for Needle-Free Transdermal Delivery Based on Their Composition

One of the most recent approaches to address the problem of needle-free vaccination is to focus on the composition of the design without losing sight of the nanometric size. The first approaches in this field were not very far from what is known today; it is necessary to have a small particle size to bypass the stratum corneum [102], and this was the predominant approach in designs for a long time. Large amounts of surfactants were used to achieve this objective. However, today it is known that it is not enough to have a small particle size and adequate lipophilicity; it is also necessary to have excipients that will allow the stratum corneum to be reversibly and non-aggressively opened and thus will enable the passage of nanosystems. Among the components that can be highlighted recently, the use of Compritol 888 ATO has been described as a promoter of transdermal penetration in polymeric nanoparticles that encapsulate ovalbumin and as adjuvant Imiquimod [7].

### 5.1. Azones

Azones and their derivatives are among the main transdermal permeability enhancers that can be used to manufacture nanosystems. These molecules are composed of a polar and a hydrophobic chain and can enhance transdermal penetration at low concentrations. It is most extensively used alongside laurocapram; however, it should be noted that this or its derivatives have not been yet incorporated for the development of nanosystems for needle-free immunization purposes [117,147].

### 5.2. Fatty Acids

Fatty acids such as oleic acid, stearic acid, and ethyl oleate are approved by the U.S. Food and Drug Administration (FDA) and can provide various advantages when incorporated into the designs. These excipients allow increasing the transdermal penetration of format dependent on its structure and chain length. The mechanism of action for the transdermal penetration increase is based on the increase in the diffusion coefficient of the skin due to the improvement of the interaction between the preparations with the lipid layer of the skin [30,124].

### 5.3. Alcohols

Alcohols are excipients that can be incorporated into nanosystems both in their internal and external phases. These can improve the solubility of encapsulated compounds and their ability to enhance the permeability of the skin. Ethanol and isopropyl alcohol have been the most used in topical preparations [148]. However, the use of alcohols represents an excellent design challenge since it is important to optimize the quantity due to their high toxicity when used in high concentrations [149,150]. Otherwise, the encapsulated compounds may be released from the nanosystems, their equilibrium may be broken, and their nanometric structure may be lost. The quantity, length of chain, and phase in which they are added suppose a significant challenge in the design [126].

### 5.4. Polymers

Polymers are excipients composed of simple subunits joined by covalent bonds called monomers and have been used in nanoparticle designs for various purposes. Among them, we can highlight the increase in the stability of the particles when they are located in external areas due to the particle-particle steric hindrance or as surfactants [108]. On the other hand, these compounds have been described in the field of nanovaccines as thermal stabilizers, and it has been shown that even at concentrations lower than those normally used as surfactants, they are capable of protecting antigens from degradation by keeping them at room temperature, without loss of efficiency [151].

### 5.5. Polysaccharides

Most polysaccharides are generally recognized as safe (GRAS), and they are widely used for various purposes. In the field of immunization, their use in formulations has been recognized for their ability to activate cells of the immune system. In the field of nanocarrier design of antigens, they stand out for their property of increasing transdermal permeability since they are capable of promoting increased permeability in the stratum corneum, which is, as has been mentioned with much emphasis, one of the main barriers to overcome [152]. These excipients have not only the property already described to enhance permeability; moreover, the incorporation of these excipients in nanoparticle-based vaccine formulations significantly increases the thermostability of antigens. An example of this has been reported for a DNA vaccine for immunization against the Ebola virus, in which the antigen in its non-encapsulated form required −70 °C for storage. In its nano encapsulated state, it no longer requires refrigeration. It is important to note that this strategy was combined with microneedles and has not been applied in needle-free systems, so there is a potential benefit with incorporation into the designs, largely because they are inexpensive excipients but with the limitation that a significant amount must be used to obtain the thermostability enhancer effect [153].

## 6. Projections

The transdermal route is a promising route of administration for immunization. Among its advantages, we find the great diversity of cells of the immune system, it is not traumatic, its great acceptance by patients, and the potential to improve the deficient characteristics of bioactive compounds. However, although multiple studies have been conducted, we do not yet have commercially available transdermal vaccines. Much progress has been made, especially in the field of microneedles, but there are still many challenges to overcome. When facing these challenges, the safety and efficacy of administering nanocomposites play a very relevant role [137]. The recent approval of parenteral vaccines that use this technology, such as the BNT162b2 vaccine (BioNTech/Pfizer) and the modern design, have shown high efficacy and safety in clinical, double-blind, multicenter, and randomized studies [154,155,156,157]. On the other hand, there is a tremendous challenge in scaling these designs to applications that can be massively developed for the population. Along these lines, it was recently shown that incorporating engineering strategies for production A batch scale of polymeric nanoparticles and even a continuous manufacturing line using microfluidics is feasible and has much potential for these applications [158].

Another relevant characteristic is the manufacturing reproducibility of the nanosystems to comply with the regulatory requirements. Therefore, the improvement of the animal study models used to evaluate the biogenicity of the materials used to constitute nanosystems needs improvement [159]. The costs associated with the production of nanocomposites will continue to be a focus of development. Undoubtedly, immunization of the population continues to be the most cost-effective way to prevent diseases [160]. Efforts continue to focus on low-cost excipients; however, only interdisciplinary work in conjunction with the production laboratories will improve production costs [161]. Needle-free administration must focus on the size and stability of the nanosystems and their composition and rational design. For years, the design and stabilization of nano-sized vehicles have relied on practical strategies and pseudo-ternary phase diagrams [134]. The most elaborate approaches have tackled the problem using Box-Behnken [162] or staggered-level designs in which various combinations of the chosen excipients are tested [163]. However, it is essential to have strategies that optimize designs in less time and consider limited resources [164]. In this context, incorporating computational techniques arises to tackle the design process with a multidisciplinary approach. The use of mathematical models and machine learning as a nanosystem optimization technique is incipient [164]. However, it is a valuable resource capable of reducing design times and the number of experiments required. Experimental data are needed to “fit” or “train” models of the effect of the composition, and they allow to explore the impact of the formulation composition in a range defined for each component [165,166].

One of the projections that this review article seeks to deliver is to motivate researchers to incorporate these types of tools that have been successfully included for the design of new and optimized promising nanosystems that not only limit their use to nanovaccines [167]. Unfortunately, the need to use elaborate programming interfaces, often away from the medical personnel who direct these investigations, has limited their use. Therefore, mathematical modeling and the use of artificial inteligence are an open door, which could allow rapid advances in the field of nanovaccines. Time and the training of interdisciplinary professionals will be able to take advantage of their potential.

## Figures and Tables

**Figure 1 vaccines-09-01420-f001:**
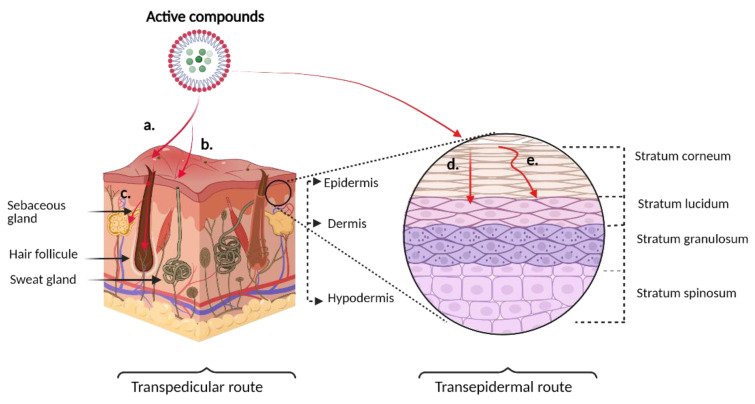
Schematic representation of the routes of skin penetration of active compounds. On the left, a transpedicular route consists of a. entry through hair follicle, b. entry through sweat glands, c. entry through sebaceous glands. On the right, transepidermal route. d. Transcellular pathway, e. Intercellular pathway.

**Figure 2 vaccines-09-01420-f002:**
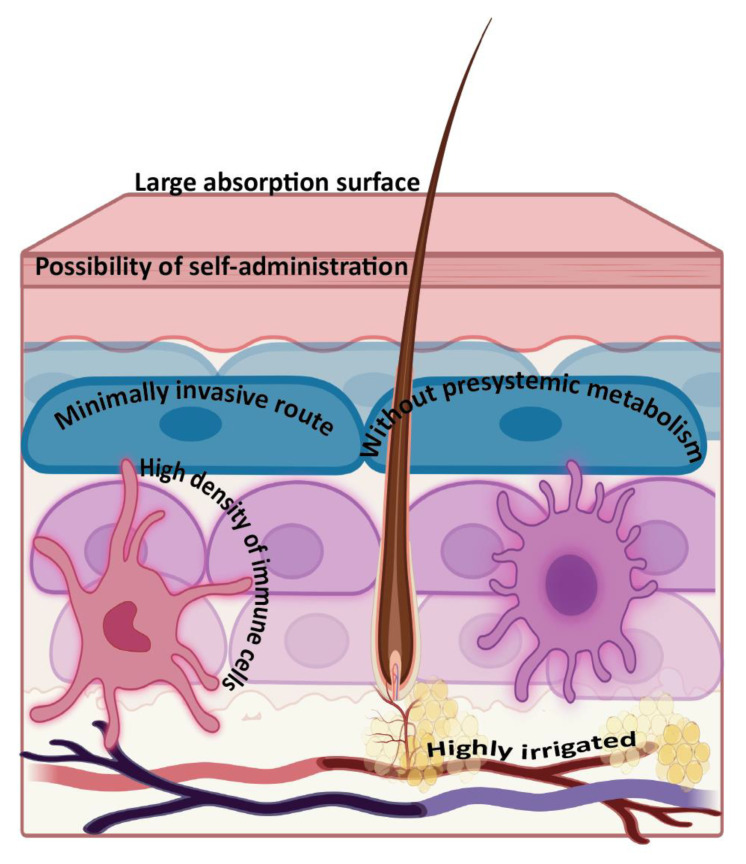
Advantages of transdermal vaccine delivery. Created with BioRender.com (accessed on 27 October 2021).

**Figure 3 vaccines-09-01420-f003:**
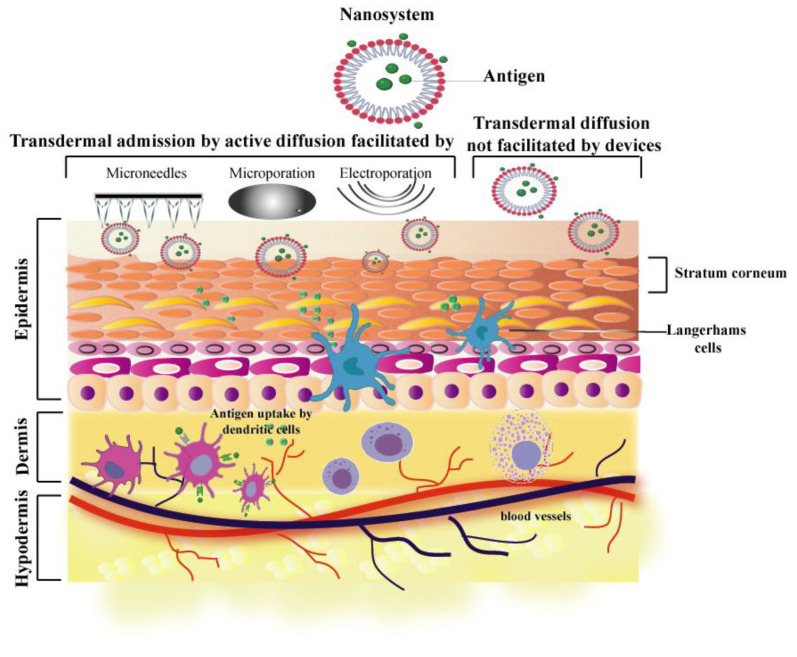
Schematic representation of the mechanisms involved in immunization based on nanoparticles, either using combined techniques or design of nanoparticles by passive diffusion. Once the stratum corneum has been crossed, the antigens can interact with cells of the immune system already described.

**Figure 4 vaccines-09-01420-f004:**
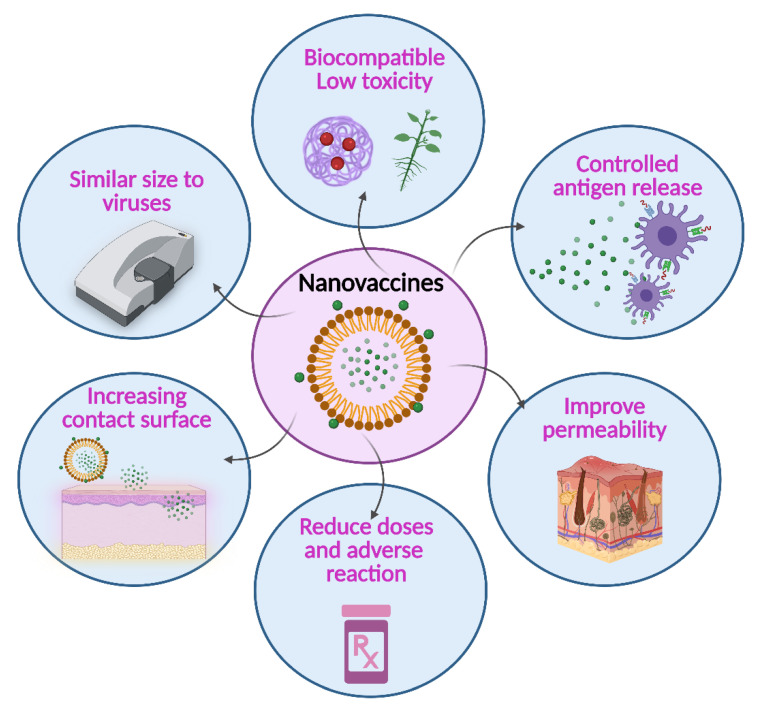
Main advantages of the use of nanocomposites for the delivery of antigens and the design of nanovaccines. Created with BioRender.com (accessed on 27 October 2021).

**Table 1 vaccines-09-01420-t001:** Transdermal nanovaccine designs for various antigens using novel nanosystems and their immune response from in vivo assays.

Antigen	Nanosystem Design	Immune Response In Vivo Assays
Ovalbumin	Liposomes, transferosomes and etosomes formulated using the reverse phase evaporation method.	In female BALB/c mice using colloidal Al (OH) 3 as adjuvant, an antiova antibody titer was obtained higher than the other nanosystems, compared to the non-encapsulated control [60].
	Phytoglycogen (PG) nanoparticles conjugated to form Nano-11 adjuvant particles with and without cyclic di-AMP, administered with Pharmajet.	The compound combining both adjuvants demonstrated a synergistic immune response that resulted in increased production of Abs IgG1 and IgG2a, as well as CD8 T lymphocytes expressing Th1, Th17 and IFN-γ in mice and pigs [61].
Homolog 5 of Plasmodium falciparum reticulocyte-binding protein (PfRH5) and coding sequence of small hepatitis B virus envelope (HBs) antigen	Tattoo Cationic liposomes fused with VHP antigen, expressing on their surface (PfRH5) formulated from dimethyldioctadecylammonium bromide) and DC-cholesterol by solvent evaporation.	A strong humoral response against PfRH5 in malaria vaccines was demonstrated in mice in those with fused tattoo, superior to the non-fused control PfRH5 and to intraperitoneal administration [62].
Diphtheria toxoid (DT)	Mesoporous Silica Nanoparticles (MSN) Embedded in Coated, Hollow Microneedles	The DT encapsulated in MSN induced a stronger antibody response than the antigen solution when administered by hollow microneedles in BALB/c mice, it is shown that the type of encapsulation and microneedle affect the response [63].
HIV-1P24-Nef peptide	PGLA nanoparticles with the sequence of the flagellin molecule from Pseudomonas aeruginosa as TLR5 activating adjuvant.	The formulation is shown in mice to improve immunogenicity and reduce the dose [64].
Antigen DNA, based on protein N or S from SARS-CoV-2 viruses	Lipidoid nanoparticles composed of low molecular weight polyethyleneimines conjugated with deoxycholic acid loaded with the adjuvant Resiquimod in separable microneedles.	The authors show in female C57BL/6 mice that the intradermal vaccine is capable of inducing an enhanced and lasting immune response compared to the intramuscular route, the formulation can be kept at room temperature for at least 30 days [65].
Influenza Neuraminidase and Flagellin Protein	Influenza 2 matrix protein ectodomain (M2e) nanoparticles (M2) by ethanol desolvation and double-layered protein nanoparticles, incorporated in soluble microneedles.	The nanovaccine was able to significantly increase the levels of specific antibodies and protect the mice from infection [66].

**Table 2 vaccines-09-01420-t002:** Main Nanocarriers used in transdermal immunization.

Nanosystem	Application in Immunization	Challenges	References
Liposomes	Microneedles combined with liposomes co-loaded with doxorubicin HCl (DOX) and celecoxib (CEL)/cationic liposomes encapsulated with hepatitis B DNA vaccine and adjuvant CpG ODN.	Conducting clinical trials, limitations associated with the coupled use of microneedles.	[81,82]
	Liposomes loaded with the surface antigen of P-falciparum MSP-1		[83,84]
	Yersinia pestis F-1 antigen-loaded liposomes using microneedles		[85]
Transferosomes	Cationic transferosomes composed of cationic lipid DOTMA and sodium deoxycholate.	Deficiency of consistent results that validate increased transdermal permeability.	[60,86]
Ethosomes	Hyaluronic acid (HA) and galactosylated chitosan (GC) modified ethosome (Eth-HA-GC) loaded ovalbumin.	Evaluation of safety and efficacy using other antigens, application suggested by authors in oncology	[87]
		High ethanolic content can be a “double-edged sword”, producing high drug entrapment, but also large leakage.	
Niosomes	Cationic niosomes loaded with ovalbumin combined with hollow microneedle.	Dependence of association with microneedles.	[88]
Cubosomes	Cubosomes that encapsulate adjuvants Quil A and monophosphoryl lipid.	Ability to cross the stratum corneum by passive diffusion, compatibility to encapsulate antigens and adjuvants in sets still under study.	[89]
	Cubosomes to transport antigens combined with microneedles		[90]
Polimeric Nanocapsules	Protamine and polyarginine nanocapsules in association with the recombinant hepatitis B surface antigen.	Incorporation of adjuvant molecules to obtain an improved immune response.	
	Nanocapsules of a vitamin E oily core, surrounded by two layers: a first layer of chitosan and a second of dextran sulphate, antigen, IutA protein from *Escherichia coli*		[91]
	Autonomous active microneedle for the direct intratumoral delivery of an immunoadjuvant, cowpea mosaic virus nanoparticles (CPMV).		[92,93,94]
	Chitosan-coated PLGA nanoparticles		

## Data Availability

Not applicable.

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
