# Peer review of "The Importance of Nanocarrier Design and Composition for an Efficient Nanoparticle-Mediated Transdermal Vaccination"

_vaccines, 2021, doi:10.3390/vaccines9121420_

Round 1

Reviewer 1 Report

This is a good review of the developmental technology associated with potential vaccine candidates. It is well developed and supported.

It would be useful to include a few more relevant details of immune responses and protection, even if only experimental. This would be especially useful to compare the relative efficiency  of overcoming dermal barriers. Additional information regarding the manufacturing processes that will be required for these formulations would be useful for the likely readers. 

The description of different types of nanoparticles and their composition is good and appears to be complete and well-supported. These descriptions are the strength of the paper.

I think there are two topics that could strengthen the paper:

1. Specific examples of immune responses with at least some of the particles comparing these responses to conventional, parenteral vaccines. This could include examples where immune responses are not optimal and what work needs to occur to resolve such questions. Most importantly, especially in laboratory animal models, are the responses protective in challenge to immunity studies? This is a vaccine paper, and it needs immunology/protection data included. The generalized descriptions of how nanoparticles interact with the immune systems is good, but specific data for types of particles would improve the paper.

2. Some information regarding how these particles could be manufactured and relative costs of goods would be useful. Are these approaches practical? I believe they are, and these data and comments would be useful. Are the nanoparticles stable? How would they be formulated for administration? Although these points are likely not addressed in final form, they are important considerations for decisions for further developmental investment.

Author Response

Answers to Reviewer 1:

Reviewer 1: This is a good review of the developmental technology associated with potential vaccine candidates. It is well developed and supported. It would be useful to include a few more relevant details of immune responses and protection, even if only experimental. This would be especially useful to compare the relative efficiency of overcoming dermal barriers. Additional information regarding the manufacturing processes that will be required for these formulations would be useful for the likely readers. The description of different types of nanoparticles and their composition is good and appears to be complete and well-supported. These descriptions are the strength of the paper.

Answer: We appreciate the positive comments about our manuscript, and hope that the current version is acceptable for publication. As requested by the Reviewer we have modified the manuscript to improve the experimental information given. We added a new table (Table 1). In this new table, we describe recent references about the use of different antigens and the immune response in in vivo assays (See table 1, in section 3.1) (lines 220 to 222).

Major comments:

Reviewer 2: Specific examples of immune responses with at least some of the particles comparing these responses to conventional, parenteral vaccines. This could include examples where immune responses are not optimal and what work needs to occur to resolve such questions. Most importantly, especially in laboratory animal models, are the responses protective in challenge to immunity studies? This is a vaccine paper, and it needs immunology/protection data included. The generalized descriptions of how nanoparticles interact with the immune systems is good, but specific data for types of particles would improve the paper.

Answer: As requested by the Reviewer, we have resolved these issues by adding a new table (Table 1). We have modified the manuscript to discuss additional experimental information regarding the recent advances in the transdermal vaccination using nanosystems .

Reviewer 3: Some information regarding how these particles could be manufactured and relative costs of goods would be useful. Are these approaches practical? I believe they are, and these data and comments would be useful. Are the nanoparticles stable? How would they be formulated for administration? Although these points are likely not addressed in final form, they are important considerations for decisions for further developmental investment.

Answer: As requested by the Reviewer, we have modified the manuscript to improve the information relative to the manufacturing process and the challenges for these methods in section 6 (lines 476 to 493).

Reviewer 2 Report

The review paper by Valdivia-Olivares et al. focuses on transdermal immunization.
Strengths:
The manuscript discusses the biological basis for transdermal immunomodulation. The researchers mentioned that the skin is an easily accessible and highly immunocompetent organ, which can have up to 20 billion cells of various subtypes, such as keratinocytes, Langerhans cells, dendritic cells, T cells, and mast cells. These cells  contribute to the immunocompetence of the skin. This is a critical area of research that continues to witness growth.
They described several types of enhancement techniques for transdermal drug delivery including iontophoresis, sonophoresis, magnetophoresis, electroporation, and laser microporation. It was pointed out that these techniques possess significant economic limitations. The authors also pointed out that microneedles have also been used to transport micro- and macromolecules through the skin. The authors made valid assertions that these technologies though beneficial also have disadvantages.
The authors also evaluated the technological advances in the area of nanomedicine. It was mentioned that nanosystems capable of transporting antigens across the skin barrier. The focus was on the most well-known systems for antigen delivery were discussed: nanoparticles, liposomes, polymeric nanoparticles, cubosomes, nanoemulsions, niosomes, ethosomes, gold and nanoparticles.
Weaknesses:
Although a topical area, most of the systems described are not novel. They have been extensively described in the literature.
Nevertheless, the review is timely, and I recommend the manuscript for publication.

Author Response

Answers to reviewer 2

Reviewer 2: The manuscript discusses the biological basis for transdermal immunomodulation. The researchers mentioned that the skin is an easily accessible and highly immunocompetent organ, which can have up to 20 billion cells of various subtypes, such as keratinocytes, Langerhans cells, dendritic cells, T cells, and mast cells. These cells contribute to the immunocompetence of the skin. This is a critical area of research that continues to witness growth. They described several types of enhancement techniques for transdermal drug delivery including iontophoresis, sonophoresis, magnetophoresis, electroporation, and laser microporation. It was pointed out that these techniques possess significant economic limitations. The authors also pointed out that microneedles have also been used to transport micro- and macromolecules through the skin. The authors made valid assertions that these technologies though beneficial also have disadvantages.

Answer: We thank the Reviewer for the positive feedback on our manuscript and hope that the current version is acceptable for publication.

Reviewer 2: The authors also evaluated the technological advances in the area of nanomedicine. It was mentioned that nanosystems capable of transporting antigens across the skin barrier. The focus was on the most well-known systems for antigen delivery were discussed: nanoparticles, liposomes, polymeric nanoparticles, cubosomes, nanoemulsions, niosomes, ethosomes, gold and nanoparticles. Weaknesses: Although a topical area, most of the systems described are not novel. They have been extensively described in the literature. Nevertheless, the review is timely, and I recommend the manuscript for publication.

Answer: As requested by the Reviewer, we have modified the manuscript to further discuss novel methodologies by adding a new table (Table 1). In this new table, we included recent information according to the use of nanosystems in different in vivo assays and the immune response for diverse antigens. Furthermore, we have improved the resolution of all figures and added a new figure (Figure 1) to show information that will facilitate understanding by readers as to how active compounds can go across the transdermal route (lines 97 to 113). Finally, we improved the spelling and grammar throughout the manuscript and added a new paragraph in section 6 that provides additional information regarding the challenges pending in this area (lines 471 to 485).

Reviewer 3 Report

The manuscript “The Importance of Nanocarrier Design and Composition for an Efficient 2 Nanoparticle-Mediated Transdermal Vaccination” is nicely written and presented. Following are suggestion which will help to further improve the review article.

Figure 2 is blurred, author should redraw it.

Authors should include the figure showing the major pathways of percutaneous absorption for nano‐sized drug carriers.

It will be better to include a table showing advantages and disadvantages of each transdermal techniques used for the delivery of vaccines.

Author should include one more column in table showing “Induced immune response”

Authors should update the information for headings: liposomes, Transferosomes, ethosomes, Niosomes, cubosomes etc…

Author Response

Answer to reviewer 3

The manuscript “The Importance of Nanocarrier Design and Composition for an Efficient 2 Nanoparticle-Mediated Transdermal Vaccination” is nicely written and presented. Following are suggestion which will help to further improve the review article.

Answer: We thank the Reviewer for the positive evaluation of our article. We hope the current version is acceptable for publication.

 Major comments: Figure 2 is blurred; author should redraw it.

Answer: As requested by the Reviewer, we have improved the visualization of the figure after inserting it in the text and the blurred elements were amended, such as nanoparticles and immune cells in the figure (now Figure 3). Further, we have modified all figures to improve sharpness and clarity.

Authors should include the figure showing the major pathways of percutaneous absorption for nano‐sized drug carriers.

Answer: As suggested by the Reviewer, we have created a new figure (Figure 1) that illustrates the entry routes of active compounds through the skin. Additionally, the text was modified to add an explanation of the percutaneous absorption of bioactive compounds (lines 97 and 114).

It will be better to include a table showing advantages and disadvantages of each transdermal techniques used for the delivery of vaccines. Author should include one more column in table showing “Induced immune response”.

Answer: As requested by the Reviewer, we have carried out a more extensive bibliographic search and created a new table (Table 1) that describes the antigens used, the nanosystems, and the immune response induced in preclinical studies as compared to the conventional administration.

Authors should update the information for headings: liposomes, Transferosomes, ethosomes, Niosomes, cubosomes etc… Answer: We appreciate the comments by the Reviewer and have added recent information to the new table to further emphasize these topics.

Answer: We appreciate the comments by the Reviewer and have added recent information to the new table to further emphasize these topics.

Reviewer 4 Report

The review article “The Importance of Nanocarrier Design and Composition for an Efficient Nanoparticle-Mediated Transdermal Vaccination” highlights the recent formulation and technological advances for transdermal delivery of vaccines. The article seems interesting to readers. However, the resolution of all figures (1, 2, and 3) are quite poor or are blurry. Thus, I suggest replacing these figures with the figures with good resolutions to improve the quality of the article.

Author Response

The review article “The Importance of Nanocarrier Design and Composition for an Efficient Nanoparticle-Mediated Transdermal Vaccination” highlights the recent formulation and technological advances for transdermal delivery of vaccines. The article seems interesting to readers. However, the resolution of all figures (1, 2, and 3) are quite poor or are blurry. Thus, I suggest replacing these figures with the figures with good resolutions to improve the quality of the article.

Answer: We greatly appreciate the positive feedback from the reviewer. As requested by the reviewer. We have reviewed all the figures, including the new ones; low-resolution elements were modified; in addition, those colors that lost sharpness with the export of the figure were changed, the pixels were increased, and the letter sharpness manager was changed. We hope that now the readers can have a correct visualization. Additionally, we have improved section 2, including a new figure (see figure 1) and the explanation about the access of bioactive compounds by the transdermal route. We improved section 3, and included a new table with recent studies and the immune response obtained for different antigens, we have added additional information in section 6, related to the challenges still pending on this topic. We hope that the current version may be appropriate for publication.

Reviewer 5 Report

This review manuscript focuses on introducing nanoparticle-mediated transdermal vaccination and the importance of nanocarrier design and composition. Due to the major barrier function of the stratum corneum layer, direct delivery of nanovaccines through the skin faces significant challenges and thus there have not been many successful examples. Successful delivery of vaccines is often achieved with effective skin disruption methods, as mentioned in the manuscript (microneedles, laser microporation, etc.). Several references were cited to show successful transdermal immunization of nanovaccines. However, authors just cited the references without elaborating more details, for example, how the vaccines were formulated, what’s the mechanisms that vaccines were successfully delivered into the skin, and how the immune responses induced by transdermal delivery of nanovaccine as compared to traditional delivery. If nanocarriers have not yet been successfully used in transdermal vaccination, there is no need to introduce. Also, the current manuscript overexaggerated the negative sides of microneedle and other skin disruption methods (e.g., skin infection risk and cost of use). When introducing skin innate immune systems, keratinocytes are not innate immune cells and can be introduced after introducing major innate immune cells of the skin (epidermal Langerhans cells and dermal macrophages and dendritic cells). Grammatical errors need to be reduced throughout.

Author Response

This review manuscript focuses on introducing nanoparticle-mediated transdermal vaccination and the importance of nanocarrier design and composition. Due to the major barrier function of the stratum corneum layer, direct delivery of nanovaccines through the skin faces significant challenges and thus there have not been many successful examples. Successful delivery of vaccines is often achieved with effective skin disruption methods, as mentioned in the manuscript (microneedles, laser microporation, etc.). Several references were cited to show successful transdermal immunization of nanovaccines. However, the authors just cited the references without elaborating more details, for example, how the vaccines were formulated, what’s the mechanisms that vaccines were successfully delivered into the skin, and how the immune responses induced by transdermal delivery of nanovaccine as compared to traditional delivery.

Answer: We appreciate the reviewer's feedback; we hope that the current improved version of the manuscript may now be appropriate for publication. As required by the reviewer. A new table has been created to strengthen this topic (See Table 1) that details the type of nanosystem used, the antigen incorporated, and preclinical tests of the immune response obtained compared to the conventional routes in the cases when was tested. A new figure has been created (see now figure 1) that allows us to understand the path that active compounds must face being administered through the skin and its corresponding explanation (see lines 97 to 114).

 If nanocarriers have not yet been successfully used in transdermal vaccination, there is no need to introduce. Also, the current manuscript overexaggerated the negative sides of microneedle and other skin disruption methods (e.g., skin infection risk and cost of use).

Answer: We appreciate the feedback provided by the reviewer. We change the redaction of this part to avoid wrong in the interpretation of the text, in this paragraph we include new information regarding the research on the skin disruption methods and the significant advances in the microneedles areas. We included a paragraph with excellent suggested revisions for the reader who needs to delve into this line of research (see lines 118 to 204).

When introducing skin innate immune systems, keratinocytes are not innate immune cells and can be introduced after introducing major innate immune cells of the skin (epidermal Langerhans cells and dermal macrophages and dendritic cells). Grammatical errors need to be reduced throughout.

Answer: We appreciate the feedback provided by the reviewer. We improve the redaction and the order in this section, we rewrote the part related to the keratinocytes and put the main immune cells at the beginning of this section (See lines 135 to 165). Additionally, we have improved the quality of all figures and incorporated new information in section 2; we have improved section 6, providing new information related to the pending challenges in this research topic.

Round 2

Reviewer 3 Report

Authors have received the manuscript as per the reviewer comments. But still there is a lot of scope to include further detail about Transferosomes, ethsosomes, niosomes, cubosomes, Nanoparticles, Nanocapsules, nanoemulsions. 

Author Response

Answer to Reviewer #3

Reviewer 3: Authors have received the manuscript as per the reviewer comments. But still there is a lot of scope to include further detail about Transferosomes, ethsosomes, niosomes, cubosomes, Nanoparticles, Nanocapsules, nanoemulsions. 

Answer: As requested by the Reviewer, we have modified the manuscript to improve section 4.  We have added and discussed recent references. Furthermore, we provide additional information that allows the reader to understand the detailed characteristics of the aforementioned nanosystems (See sub-sections from lines 334 to 481).

We would like to thank the Reviewers and the Editors for their time and effort in reviewing this work. We hope that the current manuscript is acceptable for publication in Vaccines.

Reviewer 5 Report

Authors addressed most of the comments and this version is significantly improved. Yet, some areas still need to be clarified as shown below.

  1. Drugs capable of penetrating through stratum corneum have well-defined physicochemical properties (i.e., small molecular weight less than 500 Da, p value between 1-3). The current description in line 99 and 102 is confusing since drugs need to meet multiple requirements to penetrate through the skin.
  2. Figure 1 legend a. and c. are both hair follicle routes, which are not right. Also, on the right and on the left, the left and right should be exchanged.
  3. In 2.1, only epidermal Langerhans cells are described and there is no description of dermal dendritic cells. Dermal dendritic cells need to be also introduced due to their importance in skin vaccinations.
  4. Line 193, soluble microneedles are confusing to readers.

Author Response

Answer to Reviewer #5

Reviewer 5: Authors addressed most of the comments and this version is significantly improved. Yet, some areas still need to be clarified as shown below. Drugs capable of penetrating through stratum corneum have well-defined physicochemical properties (i.e., small molecular weight less than 500 Da, p value between 1-3). The current description in line 99 and 102 is confusing since drugs need to meet multiple requirements to penetrate through the skin.

Answer: As requested by the Reviewer, we have modified the manuscript to improve the wording and clarity. Furthermore, we have included new references and new information related to the characteristics of bioactive compounds to cross the stratum corneum (See lines 93 to 99.)

Reviewer 5: Figure 1 legend a. and c. are both hair follicle routes, which are not right. Also, on the right and on the left, the left and right should be exchanged.

Answer: As requested by the Reviewer, we have modified the text to more clearly explain entry routes. Further, have explained that the letters in the figure are components and not different pathways (see lines 107 to 116). We have also corrected the wording in the figure caption, and the typing error in the panel display (see lines 131 to 135).

Reviewer 5: In 2.1, only epidermal Langerhans cells are described and there is no description of dermal dendritic cells. Dermal dendritic cells need to be also introduced due to their importance in skin vaccinations.

Answer: As requested by the Reviewer, we have added a paragraph introducing this cell line and the contribution to the immune response in the skin (see lines 163 to 167).

Reviewer 5: Line 193, soluble microneedles are confusing to readers.

Answer: As requested by the Reviewer, we have improved the wording of the referenced lines, explaining in more detail examples for soluble microneedles, to improve clarity (see lines 214 to 217).

We would like to thank the Reviewers and the Editors for their time and effort in reviewing this work. We hope that the current manuscript is acceptable for publication in Vaccines.
